# The α_2_δ Calcium Channel Subunit Accessorily and Independently Affects the Biological Function of *Ditylenchus destructor*

**DOI:** 10.3390/ijms232112999

**Published:** 2022-10-27

**Authors:** Xueling Chen, Mingwei An, Shan Ye, Zhuhong Yang, Zhong Ding

**Affiliations:** College of Plant Protection, Hunan Agricultural University, Changsha 410128, China

**Keywords:** *Ditylenchus destructor*, voltage-gated calcium channel, *DdCa_v_α_2_δ*, in situ hybridization, RNA interference

## Abstract

The α_2_δ subunit is a high-voltage activated (HVA) calcium channel (Ca_v_1 and Ca_v_2) auxiliary subunit that increases the density and function of HVA calcium channels in the plasma membrane of mammals. However, its function in plant parasitic nematodes remains unknown. In this study, we cloned the full-length cDNA sequence of the voltage-gated calcium channel (VGCC) α_2_δ subunit (named *DdCa_v_α_2_δ*) in *Ditylenchus destructor*. We found that *DdCa_v_α_2_δ* tends to be expressed in the egg stage, followed by the J3 stage. RNA-DIG in situ hybridization experiments showed that the *DdCa_v_α_2_δ* subunit was expressed in the body wall, esophageal gland, uterus, post uterine, and spicules of *D. destructor*. The in vitro application of RNA interference (RNAi) affected the motility, reproduction, chemotaxis, stylet thrusting, and protein secretion of *D. destructor* to different degrees by targeting *DdCα1D*, *DdCα1A*, and *DdCa_v_α_2_δ* in J3 stages*,* respectively. Based on the results of RNAi experiments, it was hypothesized that L-type VGCC may affect the motility, chemotaxis, and stylet thrusting of *D. destructor*. Non-L-type VGCC may affect the protein secretion and reproduction of *D. destructor*. The *DdCa_v_α_2_δ* subunit gene also affected the motility, chemotaxis, and reproduction of *D. destructor*. These findings reveal the independent function of the VGCC α_2_δ subunit in *D. destructor* as well as give a theoretical foundation for future research on plant parasitic nematode VGCC.

## 1. Introduction

Intracellular Ca^2+^ influx is primarily mediated by voltage-gated calcium channels (VGCC), which are widely distributed in biological cell membranes. Based on electrophysiological and pharmacological characteristics, VGCCs are classified as L-type, non-L-type, or T-type calcium channels [1]. Based on their activation properties, they are divided into high-voltage activated (HVA) calcium channels and low-voltage activated (LVA) calcium channels, with the HVA calcium channel being divided into L-type (Ca_V_1) and non-L-type (Ca_V_2) channels, whereas the LVA calcium channel only has T-type channels (Ca_V_3) [2]. In mammals, calcium channels are found in a variety of cells, including neurons, neurosecretory cells, and muscle cells, and they play an important role in muscle contraction, hormone secretion, and neurotransmitter release. Ca_V_1 and Ca_V_2 channels are typically made up of a pore-forming α_1_ subunit and β, α_2_δ, and γ auxiliary subunits, whereas only an α_1_ subunit has been identified in Ca_V_3 channels [3,4,5].

Although the major biophysical and pharmacological features of these channels are determined by the α_1_ subunit, their expression is also influenced by auxiliary subunits β and α_2_δ. The β subunit is one of the major auxiliary subunits of HVA calcium channels, and four of its genes (β1–β4) have been cloned. The GK domain of the β subunit binds to the α-interacting domain (AID), thereby exerting its role in regulating the surface expression and gating properties of high-voltage activated calcium channels [6,7,8]. α_2_δ is another major auxiliary subunit of the HVA calcium channel, which increases the density of these channels in the plasma membrane, thereby enhancing their function [9,10]. Four α_2_δ subunit genes in vertebrates have been cloned: CACNA2D1, CACNA2D2, CACNA2D3, and CACNA2D4 [9,11,12,13]. 

α_2_δ consists of two subunits, α_2_ and δ, and both subunits are encoded by a single gene and covalently linked by disulfide bonds. Two cysteine (Cys) residues, Cys404 (located in the Von Willebrand Factor A domain (VWA domain) of α_2_) and Cys1047 (located within the extracellular domain of δ) in α_2_δ-1, are involved in the formation of intermolecular disulfide bonds [14]. α_2_ is completely extracellular, but there is controversy about the transmembrane mode of δ. It was initially suggested that δ belongs to the type I transmembrane proteins, while an increasing number of findings suggest that δ may be a glycosylphosphatidylinositol (GPI)-anchored protein. Davies et al. [15] proposed that α_2_δ-1 is a GPI-anchored protein and suggested that GPI anchoring of the α_2_δ subunit is required for its enhanced calcium current. Similarly, both the α_2_δ-3 and α_2_δ-4 genes were shown to be GPI-anchored proteins in mice, with the predicted GPI-anchoring motifs being CGG or GAS [16,17]. There is no doubt that all α_2_δ contain the VWA and Cache domains. The VWA domain contains ~200 amino acids and represents a dinucleotide binding fold with a metal ion-dependent adhesion site (MIDAS) motif, which participates in divalent cation-dependent interactions, and VWA structural domains are involved in protein–protein interactions through the MIDAS motif [18]. 

The MIDAS motif of α_2_δ-1 and α_2_δ-2 is “perfect”, and its key component is a five-residue motif with three ligand residues (D × S × S) near the N terminus of the VWA domain [19], which is also present in α_2_δ-3 and α_2_δ- 4 [20]. Downstream of the VWA structural domain, the α_2_δ subunit also has a bacterial chemosensory structural domain called the Cache domain [21]. The Cache domain of the α_2_δ subunit may be involved in the transport function of the α_2_δ subunit. α_2_δ-1 and α_2_δ-2 are target sites for gabapentin-like drugs [22], and it has been hypothesized that the Cache domain is associated with the binding of gabapentin drugs and with the binding of the putative endogenous ligand [23,24]. UNC-36 and TAG-180 are the α_2_δ subunits of VGCC from *Caenorhabditis elegans* [25]. UNC-36 has been shown to affect the voltage dependence, kinetics, and conductance of voltage-dependent calcium currents and to play a key role in striated muscle, whereas TAG-180 does not [26]. UNC-36 also plays a central role in the excitability and the functional activity of *C. elegans* mechanosensory neurons [27].

The sweet potato rot nematode, *Ditylenchus destructor*, is a migratory endoparasitic nematode from clade 12 based on a small subunit ribosomal DNA (SSU rDNA) sequence tree [28]. Moreover, *D. destructor* is a polyphagous worm with a wide host range that can live on fungi and plants. It can harm a range of plants, including peanuts, carrots, and *Codonopsis pilosula,* in addition to its two primary hosts of sweet potato and potato [29,30,31]. Furthermore, *D. destructor* may eat a variety of fungi [32]. *Ditylenchus destructor* became a quarantine focus in various nations due to the massive agricultural losses it has caused.

VGCC have been well studied in mammals, but few studies have been conducted in plant parasitic nematodes. We previously identified the L-type (*Dd**Cα1D*), non-L-type (*Dd**Cα1A*), and T-type (*Dd**Ca1G*) VGCC α_1_ subunit genes in *D. destructor* and showed that *Dd**Cα1D* is expressed in the body wall muscles of *D. destructor* and affects its motility, whereas *Dd**Cα1A* is expressed in the esophageal glands, vulva, and vas deferens of *D. destructor* and affects its reproduction [33]. Here, we successfully cloned the gene for the VGCC α_2_δ subunit from *D. destructor* (*DdCa_v_α_2_δ*), determined its position using in situ hybridization, and characterized its biological properties and functions using RNA interference techniques. We demonstrated that silencing the *DdCα1D* gene affects the motility, chemotaxis, and stylet thrusting of *D. destructor*, and silencing the *DdCα1A* gene affects *D. destructor*’s protein secretion and reproduction. Moreover, we showed that the auxiliary subunit *DdCa_v_α_2_δ* gene contributes to these biological functions of the VGCC α_1_ subunit. 

Consequently, the silencing of the *DdCa_v_α_2_δ* subunit gene alone also affected the motility, chemotaxis, and reproduction coefficient of *D. destructor***.**

## 2. Results

### 2.1. Cloning and Characterization of DdCa_v_α_2_δ

RT-PCR was conducted using specific primers (Table 1), and PCR products were sequenced to confirm the amplification of the full-length cDNA sequence from *D. destructor*. *DdCa_v_α_2_δ* included a 3825 bp open reading frame (ORF), identified with the NCBI ORF finder, that encoded 1275 amino acids (GenBank accession number MW267435) with a molecular mass of 147 kDa and a pI of 6.67. *DdCa_v_α_2_δ* began with an ATG initiation codon following the 312-bp upstream 5′ untranslated region (UTR) and ended with a TGA stop codon upstream of the 635-bp 3′ UTR (Figure 1). SL1 is a guide and a widely conserved sequence present at the 5’ end of the mRNA of most nematodes [34], and the presence of 5′ trans-spliced leader sequences in the nematode phylum allows the use of an SL1-PCR strategy to clone full-length cDNAs from very small amounts of RNA [35]. In our study, the sequence obtained by SL1-PCR and 3′RACE has a start codon (ATG) upstream of the coding framework, a stop code (TGA) downstream, and a polyA tail at its 3′ end. Therefore, we indicated that this is the complete cDNA sequence.

There are 22 cysteine (Cys) residues in the *DdCa_v_α_2_δ* amino acid sequence and nine predicted N-glycosylation sites (Figure 2). These cysteine residues may be associated with the formation of disulfide bonds. The amino acid sequence of *DdCa_v_α_2_δ* was compared with the unc-36 sequence from *C. elegans* and four α_2_δ sequences from humans. BLAST Protein analysis with the encoded amino acid sequences showed that *DdCa_v_α_2_δ* has features in common with other VGCC α_2_δ subunits, including two typical regions: the VWA domain and the Cache domain (Figure 3). Moreover, according to Qin et al., we also predicted the breaking point between α_2_ and δ subunits in the *DdCa_v_α_2_δ* sequence (Figure 3) [12]. VWA domains are usually about 200 residues long, and there is a MIDAS motif in each VWA domain. We found that the VWA domains here all contained a “perfect” MIDAS motif, in which all five discontinuous and co-coordinating amino acids were present (DxSxS, A/T, D) [18]. The amino acid position of the VWA domain of the *DdCa_v_α_2_δ* sequence was present at 270–487 amino acids (Figure 1), and this domain was highly conserved when compared with the unc-36 domain, with a similarity of 81.3%. The Cache domains of *DdCa_v_α_2_δ*, unc-36, and tag-180 were all downstream of the VWA domains. The amino acid positions of the Cache domains in the *Dd*Cavα_2_δ sequence was 519–602 amino acids. The MIDAS motif in the *DdCa_v_α_2_δ* sequence was highly conserved in comparison to the unc-36 sequence. These sequences all contained five co-coordinating amino acids, including D, S, S, A, and D. 

### 2.2. Phylogenetic Analysis of DdCa_v_α_2_δ

The phylogenetic analysis was based on the deduced amino acid sequences of the α_2_δ subunits of *D. destructor* and the corresponding subunits of *C. elegans* and other species. The amino acid sequences were obtained from the NCBI (https://www.ncbi.nlm.nih.gov/, accessed on 15 October 2022) and WormBase (https://parasite.wormbase.org/index.html, accessed on 15 October 2022) databases (Table 2). The phylogenetic tree shows the branches composed of vertebrate and invertebrate subunits. Particularly, the α_2_δ of *D. destructor* and the corresponding subunits of *C. elegans* were clustered in invertebrates, forming a different branch from vertebrates, and they were closely related to the evolutionary position of *C. elegans* (Figure 4). This implies that *DdCa_v_α_2_δ* may have the same evolutionary relationship and a similar physiological function as the *unc-36* gene of *C. elegans*.

### 2.3. Developmental Expression Analysis of DdCa_v_α_2_δ

The expression level of *DdCa_v_α_2_δ* was assessed by RT-qPCR at different developmental stages (including egg, J2, J3, and J4). *DdCa_v_α_2_δ* was widely expressed at all developmental stages (Figure 5). Results indicated that *DdCa_v_α_2_δ* plays an important role in all stages of *D. destructor* development. In particular, the expression of *DdCa_v_α_2_δ* was significantly higher in the egg stage than in other developmental stages. In addition, the mRNA expression level of *DdCa_v_α_2_δ* in the J3 and J4 were 3.4 and 2.7 times higher than in the J2 stage, respectively.

### 2.4. Tissue Localization of DdCa_v_α_2_δ

In situ hybridization was used to detect the site of *DdCa_v_α_2_δ* gene expression in the adult stages of *D. destructor* at the mRNA level (Figure 6). The antisense-probe treated nematodes showed positive staining in the esophageal glands (Figure 6E), body wall (Figure 6F), uterus and post uterine area of females (Figure 6G), and the spicules of males (Figure 6H), whereas the sense-probe treated nematodes showed no hybridization signal (Figure 6A–D).

### 2.5. Silencing Efficiency of dsRNA Soaking

Following 24 h of dsRNA treatment, FITC was taken up by *D. destructor*. The expression levels of *DdCα1D**, DdCα1A*, and *DdCa_v_α_2_δ* mRNA in *D. destructor* were measured by qPCR. When ds*Ca_v_α_2_δ* was fed, the transcript level of *DdCa_v_α_2_δ* was significantly reduced to 31.4%, whereas the transcript levels of *DdCα1D* and *DdCα1A* were unchanged relative to the control (Figure 7A). When ds*Cα1D* was fed, the transcript level of *DdCα1D* was significantly reduced to 67.8%, but the transcript levels of *DdCα1A* and *DdCa_v_α_2_δ* were unchanged (Figure 7B). When ds*Ca_v_α_2_δ* and ds*Cα1D* were fed simultaneously, the transcript levels of *DdCa_v_α_2_δ* and *DdCα1D* decreased to 45.7% and 67.4%, respectively, whereas the transcript level of *DdCα1A* was unchanged (Figure 7D). When ds*Cα1A* was fed, the transcript level of *DdCα1A* decreased to 44.7%, whereas there was no change in the transcript levels of *DdCα1D* and *DdCa_v_α_2_δ* (Figure 7C). When ds*Ca_v_α_2_δ* and ds*Cα1A* were fed simultaneously, the transcript levels of *DdCa_v_α_2_δ* and *DdCα1A* were decreased to 41.2% and 33.3%, respectively, and the transcript level *DdCα1D* was unchanged (Figure 7E). 

The above results showed that, when the dsRNA for each gene was fed, this led to a significant specific partial knock down of the target gene, indicating that our RNAi approach works. Furthermore, silencing of the *DdCa_v_α_2_δ* subunit gene did not lead to the downregulation of the transcript levels of the *DdCα1D* and *DdCα1A* subunit genes. Silencing of the *DdCα1D* or *DdCα1A* subunits also did not lead to the downregulation of the *DdCa_v_α_2_δ* subunit gene at the transcript level, indicating that the expression of *DdCa_v_α_2_δ* does not affect the transcription of the *DdCα1D* and *DdCα1A* subunits. Thus, it is presumed that the α_2_δ and HVA α_1_ subunits do not appear to influence each other’s mRNA expression levels. 

### 2.6. Phenotypic Analysis of DdCα1D, DdCα1A, and DdCa_v_α_2_δ Genes after Knockout

#### 2.6.1. Motility Assay

Results of the motility assay are shown in Figure 8. After washing dsRNA for 6 h, when ds*GFP* was fed, its passage rate was 33.7%. The sand column passage rates of other target dsRNA genes were significantly downregulated compared with ds*GFP* treatment. When ds*Ca_v_α_2_δ* was fed, its passage rate was 25.3%. When ds*Ca1D* was fed, its passage rate was 23.7%. When ds*Ca_v_α_2_δ* and ds*Cα1D* were fed simultaneously, the passage rate was 19.7%. When ds*Ca1A* was fed, its passage rate (28.3%) was higher than that of ds*Ca1D*. When ds*Ca_v_α_2_δ* and ds*Cα1A* were fed simultaneously, the passage rate (25.7%) was higher than that of ds*Ca_v_α_2_δ* + ds*Cα1D*. After washing dsRNA for 24 h, the sand column passage rate of *D. destructor* in all treatments increased significantly, but there were still significant differences compared to that observed in the ds*GFP* treatment. 

The passage rate of ds*GFP* treatment was 54.3%. When ds*Ca_v_α_2_δ* was fed, the passage rate of *D. destructor* was 44.7%, and the passage rate was 42.0% when ds*Ca1D* was fed. In addition, the nematode passage rate was 41.0% when ds*Ca_v_α_2_δ* and ds*Ca1D* were fed simultaneously. The passage rate was 46.7% for the nematodes fed ds*Ca1A* and 48% for nematodes fed both ds*Ca_v_α_2_δ* and ds*Cα1A*. The above results indicated that the effect of RNA interference diminished the motility of nematodes, and this decrease in motility recovered with time. Additionally, results indicate that both L-type and non-L-type VGCC affect the motility of *D. destructor*, and the Ca_v_α_2_δ subunit affects *D. destructor* locomotion. The Ca_v_α_2_δ subunit might play an auxiliary role in the locomotion of *D. destructor* that is mediated by L-type and non-L-type VGCCs. 

#### 2.6.2. Chemotaxis Assay

In order to detect changes in *D. destructor* chemotaxis to sweet potato after silencing the *DdCa_v_α_2_δ* subunit gene, the number of nematodes attracted to sweet potato blocks placed on 1% agar for 36 h was counted separately. The results are shown in Figure 9A. The attraction rate of the ds*GFP* treatment was 22.5%. Compared with the ds*GFP* treatment, the nematode attraction rates of the other target gene dsRNA treatments were decreased to various degrees. When ds*Ca1D* was fed, the attraction rate of *D. destructor* was 7.3%. When ds*Ca_v_α_2_δ* was fed, the attraction rate was 12.0%. When ds*Ca_v_α_2_δ* and ds*Ca1D* were fed simultaneously, the attraction rate was only 5.7%. The attraction rate was 12.7% when ds*Ca1A* was fed and 9.0% for nematodes fed both ds*Ca_v_α_2_δ* and ds*Cα1A*. The co-silencing of *DdCα1D* or *DdCα1A* and *DdCa_v_α_2_δ* subunit genes also affected nematode chemotaxis, with the effect of ds*Ca_v_α_2_δ* + ds*Cα1D* treatment being greater. This indicates that both L-type and non-L-type VGCC affect the chemotaxis of *D. destructor*, and the Ca_v_α_2_δ subunit has an important auxiliary role in L-type and non-L-type VGCC-mediated nematode chemotaxis. The Ca_v_α_2_δ subunit also had an effect on the chemotaxis of *D. destructor*.

#### 2.6.3. Stylet Thrusting Assay

To assess the effect of HVA α_1_ and α_2_δ subunits on nematode stylet thrusting, nematodes were stimulated with serotonin, and the numb of stylets thrusting was counted over a one-minute period. It is evident from Figure 9B that the number of stylets thrusting following ds*GFP* treatment was 51.7 times/min. The number of stylets thrusting in ds*Ca_v_α_2_δ*, ds*Ca1A,* and ds*Ca_v_α_2_δ* + ds*Cα1A* treatments were not significantly different from that of ds*GFP* treatment: ds*Ca_v_α_2_δ* treatment nematodes had 49.7 stylets thrusting in one minute, ds*Ca1A* treatment nematodes had 42.0 stylets thrusting, and ds*Ca_v_α_2_δ* + ds*Cα1A* treatment nematodes had 41.0 stylets thrusting. The ds*Ca1D* and ds*Ca_v_α_2_δ* + ds*Cα1D* treatment nematodes showed significant differences in the number of stylets thrusting compared to that observed in the ds*GFP* treatment. The number of nematode stylets thrusting was 33.0 times/min in the ds*Ca1D* treatment, and 27.3 times/min in the ds*Ca_v_α_2_δ* + ds*Cα1D* treatment. These results indicate that the silencing of the *DdCα1D* subunit gene alone affects the stylets thrusting of *D. destructor*, and the co-silencing of the *DdCα1D* and *DdCa_v_α_2_δ* subunit genes increases the effect of L-type VGCCs on the function of stylets thrusting. However, the difference between ds*Cα1D* and ds*Ca_v_α_2_δ* + ds*Cα1D* treatments was not significant. This indicates that L-type VGCCs affect the stylet thrusting function of the potato decay stem nematode, and non-L-type VGCCs have no effect on the stylet thrusting function of the potato decay stem nematode. Consequently, the auxiliary subunit Ca_v_α_2_δ has an important auxiliary effect on L-type VGCC-mediated stylet thrusting in *D. destructor*, but non-L-type VGCCs have no direct effect on the stylet thrusting function of *D. destructor*.

#### 2.6.4. Protein Secretion Assay

Secreted proteins from nematodes were detected after 16 h of 0.1% resorcinol treatment. As shown in Figure 9C, the nematode secretory protein content in the ds*GFP* treatment was 20,222.22 μg/mL. When ds*Ca_v_α_2_δ* and ds*Cα1D* were fed, respectively, the protein content of *D. destructor* was not significantly different compared to that of the ds*GFP* treatment, which were 20,042.22 μg/mL and 20,004.44 μg/mL, respectively. Similarly, the protein content of *D. destructor* was 20.102 mg/mL and was not significantly different compared to that in the ds*GFP* treatment when ds*Ca_v_α_2_δ* and ds*Cα1D* were fed simultaneously. However, the protein content was significantly downregulated when ds*Cα1A* or ds*Ca_v_α_2_δ* + ds*Cα1A* were fed, resulting in a protein secretion content of 19,740.00 μg/mL and 19,664.44 μg/mL, respectively. These results indicate that silencing the *DdCα1A* subunit gene alone affects protein secretion in potato rot stem nematodes and that co-silencing the *DdCα1A* and *DdCa_v_α_2_δ* subunits increases the effect of non-L-type VGCCs on the protein secretion function of *D. destructor*. This indicates that L-type VGCCs have no effect on protein secretion function, but non-L-type VGCCs affect the protein secretion function of *D. destructor*. The *Ca_v_α_2_δ* auxiliary subunit has an important auxiliary effect on nematode protein secretion mediated by non-L-type VGCCs, but it has no effect on the protein secretion function of *D. destructor*.

#### 2.6.5. Reproduction Assay

The reproduction rate was significantly reduced 25 d after *DdCα1D*, *DdCα1A*, and *DdCa_v_α_2_δ* were silenced. As shown in Figure 9D, the reproduction coefficient of *D. destructor* was 62.3 in the ds*GFP* treatment. The reproduction coefficients were significantly reduced 25 d after *DdCα1D*, *DdCα1A,* and *DdCa_v_α_2_δ* were silenced. The reproduction coefficient was 35.2 for the ds*Ca_v_α_2_δ* treatment, 32.4 for the ds*Ca1D* treatment, and 31.5 for the ds*Ca_v_α_2_δ* + ds*Cα1D* treatment. When ds*Ca1A* was fed, the reproduction coefficient was 22.2, and the reproduction coefficient was 17.2 when ds*Ca_v_α_2_δ* and ds*Cα1A* were both fed. The above results indicate that the silencing of *DdCα1D*, *DdCα1A,* and *DdCa_v_α_2_δ* subunit genes, respectively, all affected the reproduction coefficient of potato rot stem nematodes, and the silencing of the *DdCα1A* subunit gene had the greatest effect on the reproduction coefficient of nematodes. Co-silencing *DdCα1D* or *DdCα1A* with *DdCa_v_α_2_δ* subunit genes further reduced the reproduction coefficient of *D. destructor*, which was even lower in the ds*Ca_v_α_2_δ* + ds*Cα1A* treatment. These data indicate that both L-type and non-L-type VGCCs affect the reproduction of *D. destructor*, and the Ca_v_α_2_δ auxiliary subunit has an important auxiliary role in non-L-type VGCC-mediated nematode reproduction. The Ca_v_α_2_δ subunit also has an effect on the reproduction of nematodes. Among the Ca_v_α_2_δ subunits, non-L type VGCCs had the greatest effect on the reproduction of *D. destructor*.

## 3. Discussion

In mammals and insects, voltage-gated sodium channels (VGSCs) play an important role in maintaining cellular excitability and normal physiological functions, making them important targets for a variety of neurotoxins [36]. However, VGSCs have not been found in nematodes [37], whose neuronal activity is thought to be related to VGCCs. VGCCs regulate a number of physiological functions, including neuronal excitability, transmitter release, and muscle contraction and are mainly composed of several subunits, such as α_1_, β, α_2_δ with γ [38]. In our study, we cloned and characterized a VGCC α_2_δ subunit from *D. destructor*, named *DdCa_v_α_2_δ*. α_2_δ is a highly glycosylated extracellular protein containing a VWA domain that is normally found in extracellular matrix proteins and integrin receptors [39]. Therefore, it is highly likely that the interaction between α_2_δ and α_1_ occurs extracellularly. The amino acid sequence deduced from the *DdCa_v_α_2_δ* cDNA sequence has a VWA domain and a MIDAS motif, and the domain and motif are also present in the unc-36 gene of *C. elegans* [19]. Thus, we tentatively defined *DdCa_v_α_2_δ* as the VGCC auxiliary subunit of *D. destructor*. The analysis of the protein amino acid sequence also revealed that there was no GPI anchor site, “CGG” or “GAS”, as found in mammals, but a similar “GCS” sequence was found at the 903–905 amino acid position (underlined part of Figure 2). Therefore, further validation of the *DdCa_v_α_2_δ* structure that spans the membrane remains to be conducted.

Four α_2_δ subunits (α_2_δ-1, α_2_δ-2, α_2_δ-3, and α_2_δ-4) have been identified in mammals, and Dolphin et al. showed that they are expressed in skeletal muscle, neurons, the brain, and the testis [9,20]. Two VGCC α_2_δ subunits (UNC-36 and TAG-180) were identified in *C. elegans* [25]. It was found that UNC-36 was expressed in the body wall, vulva, and pharynx muscles of *C. elegans* [27]. In a further study, UNC-36 was also shown to be expressed in muscle and motor neurons and co-regulated with EGL-19 (L-type α_1_) in *C. elegans* body muscles [26,27]. Ye et al. cloned three VGCC α_1_ subunit genes in *D. destructor*, *DdCα1D*, *DdCα1A*, and *DdCa1G*, and found that they play a role in modulating locomotion, feeding, and reproduction, respectively [33]. Recently, we identified the *DdCa_v_β* subunit of *D. destructor* and showed that it has a complementary role in the biological functions of the *DdCα1D* and *DdCα1A* subunits [40]. 

In the present study, *DdCa_v_α_2_δ* was found to be expressed in the esophageal glands, body wall, uterus, and post uterine tissue of *D. destructor,* along with the spicules. This is highly consistent with studies on α_2_δ subunits in mammals and in *C. elegans.* Furthermore, we also showed that silencing the *DdCα1D*, *DdCα1A*, and *DdCa_v_α_2_δ* subunit genes, respectively, significantly reduced the nematode passage rate in sand columns and the attraction rate of *D. destructor*, with lower passage and attraction rates when the *DdCα1D* subunit gene was silenced alone. In addition, the passage and attraction rates were lowest when *DdCa_v_α_2_δ* was co-silenced with the *DdCα1D* subunit, but the rates were not significantly different from those observed with *DdCα1D* subunit gene silencing alone. The results showed that silencing of the *DdCα1D* subunit gene affected the stylet thrusting of *D. destructor*, and the co-silencing of the *DdCα1D* and *DdCa_v_α_2_δ* subunit genes increased the effect of L-type VGCCs on the stylet thrusting of nematodes, but the difference was not significant.

By examining the protein content in the supernatant of *D. destructor*, we found that silencing the *DdCα1A* subunit gene reduced the nematode’s secretory protein content, and co-silencing of *DdCa_v_α_2_δ* with the *DdCα1A* subunit gene increased the effect of non-L-type VGCCs on the nematode’s secretory protein function. Interestingly, we found some differences in secretory protein content when *DdCa_v_α_2_δ* or *DdCa_v_β* were co-silenced with the *DdCα1A* subunit gene [40]. This may be due to the fact that the Ca_v_β and Ca_v_α_2_δ auxiliary subunits have different affinities for the HVA Ca_v_α_1_ subunit, thus causing a difference in the auxiliary effect on the α_1_ subunit. The binding of the Ca_v_β to Ca_v_α_1_ subunits occurs at a high affinity action site [41], while the affinities between Ca_v_α_1_ and Ca_v_α_2_δ, however, appear rather weak [42]. We also found that the silencing of *DdCα1D*, *DdCα1A,* and *DdCa_v_α_2_δ* subunit genes, respectively, all affected the reproduction coefficient of *D. destructor*, and the silencing of the *DdCα1A* subunit gene had the greatest effect on the reproduction coefficient. Co-silencing of *DdCα1D* or *DdCα1A* and *DdCa_v_α_2_δ* subunit genes further reduced the reproduction coefficient, whereas the reproduction coefficient of ds*Ca_v_α_2_δ* + ds*Cα1A* treated nematodes was even lower. 

In this study, we found that *DdCa_v_α_2_δ* was expressed in the uterus, post uterine tissue, and spicules. *DdCα1A* was also expressed in the vulva and vas deferens [33]. Therefore, we conclude that *DdCα1A* plays a key role in the reproduction of *D. destructor* along with *DdCa_v_α_2_δ*. In the future, additional studies on the phenotypic effects of transgenic fungi on *D. destructor* should be conducted. This will provide a theoretical basis for control strategies against *D. destructor*.

## 4. Materials and Methods

### 4.1. Nematode

*Ditylenchus destructor* was isolated from infested sweet potatoes in Hebei, China, and was preserved in the storage roots of sweet potatoes. Sweet potatoes were washed with clean water, sterilized with 1% NaClO for 10 min, dried, and treated with UV light for 30 min. Approximately 1000 mixed stage *D. destructor* were inoculated into sweet potatoes by digging holes with a sterile hole punch, after which *D. destructor* were sealed in the sweet potato with paraffin. Inoculated sweet potatoes were incubated at 25 °C for 25–30 d in the dark, and nematodes in the mixed stage were collected using the modified Baermann method [43]. Nematode eggs were screened by density-gradient centrifugation with a 1500-mesh sieve [44]. Nematodes at different developmental stages were obtained at 1-week intervals. 

### 4.2. Cloning of the DdCa_v_α_2_δ Subunit

Total RNA was extracted from *D. destructor* using Trizol Reagent (Sangon Biotech Co., Ltd., Shanghai, China). RNA quality and concentration were determined with an ultra-micro spectrophotometer (Thermo, Shanghai, China), and RNA integrity was assessed using 1% gel electrophoresis. The first strand of cDNA was synthesized using the PrimeScript™Ⅲ First-Strand Synthesis System (Invitrogen, Carlsbad, USA) for RT-PCR. The *DdCa_v_α_2_δ* gene putative sequence was obtained from the WormBase database (https://parasite.wormbase.org/index.html, accessed on 15 October 2022) by comparing WormBase database genes with the *C. elegans* unc-36 gene obtained from the National Center for Biotechnology Information (NCBI) database. First, to obtain the full length *DdCavα_2_δ* subunit coding region sequence, PCR was performed using gene-specific primers (D-F, D-R) (Table 1). PCR was conducted using a standard procedure in a 25-μL volume that included 2.5 μL 10 × Ex Taq Buffer (TaKaRa, Japan), 2 μL dNTP Mixture (TaKaRa), 1 μL cDNA template, 1 μL of each primer, 0.5 μL of EX Taq polymerase (TaKaRa), and 17 μL of ddH_2_O. PCR conditions included an initial denaturation step at 94 °C for 5 min, followed by 35 cycles at 94 °C for 30 s, 55 °C for 30 s, 72 °C for 4 min, and a final step for 7 min at 72 °C. PCR products were purified using an agarose gel recovery kit (Trans, Beijing, China). The purified fragments were cloned into pMD™ 19-T vectors (TaKaRa,) and transformed into *E. coli* DH5α competent cells (Tiangen, Beijing, China). Three positive clones were randomly selected for bidirectional sequencing through the commercial service of the Sangon Biotech Co., Ltd. To obtain the length of the *DdCa_v_α_2_δ* subunit with its 5′-noncoding region sequence, PCR was performed using a spliced leader sequence, SL1, and a gene-specific primer (5′-R1, 5′-R2) (Table 1). Finally, the first strand of cDNA with the 3′-noncoding region was synthesized using the 3′ full RACE Core Set with PrimerScript^TM^RTase (TaKaRa) with the primers listed in Table 1.

### 4.3. Gene Characterization and Phylogenetic Analysis

Three overlapping fragments were spliced using DNAMAN software (DNAMAN 9.1; Lynnon BioSoft, Canada) to generate the full-length cDNA for the *DdCa_v_α_2_δ* subunit, and the amino acid sequence was deduced. The *DdCa_v_α_2_δ* conserved structural domains were predicted by the NCBI database (https://blast.ncbi.nlm.nih.gov/Blast.cgi, accessed on 15 October 2022) and were mapped using IBS 1.0. A phylogenetic tree of the *DdCa_v_α_2_δ* subunit was constructed using the mega 6.0 neighbor joining method [45]. 

### 4.4. Stage-Specific Expression of DdCa_v_α_2_δ

To analyze the expression of the *DdCa_v_α_2_δ* subunit at each developmental stage in *D. destructor*, mRNA was extracted from approximately 1000 worms using the Dynabeads^TM^ mRNA DIRECT^TM^ Kit (Invitrogen). Additionally, cDNA was synthesized from mRNA using reverse transcription kits, according to the manufacturer’s instructions. Primers were designed (Table 1), and SYBR Green Real-time PCR was performed using the 18S rRNA as an internal reference. The qPCR data were analyzed using the CFX Manager software and the Bio-Rad 2^−ΔΔCt^ method to calculate relative gene expression [46]. All experiments were repeated three times, and averages were calculated. 

### 4.5. Tissue Localization of DdCa_v_α_2_δ

In situ hybridization was performed following a previously published method with modifications [47]. Fragments used as probes were amplified from the full-length cDNA from *D. destructor* using HindIIIF and EcoRI R primers (Table 1). DIG-labeled forward and antisense probes were synthesized using the DIG RNA Labeling Kit (Roche, Basel, Germany). Approximately 10,000 mixed-stage *D. destructor* specimens were fixed in 4% Paraformaldehyde Fix Solution (Sangon Biotech Co., Ltd.) for 18 h at 4 °C, followed by fixation for 4 h at room temperature. *Ditylenchus destructor* were then cut into 2–5 segments and treated with 0.5 mg/mL proteinase K at room temperature for 1 h. Hybridization was performed at 50 °C for 22 h, and they were treated with an antibody solution at 37 °C for 2 h. Samples were treated with chromogenic solution (Sangon Biotech Co., Ltd.) overnight and then observed and photographed under a microscope (Motic, Xiamen, China).

### 4.6. dsRNA Synthesis and Soaking of D. destructor

Total RNA was extracted from *D. destructor* using the Trizol method, first strand cDNA was synthesized, and dsRNA was synthesized using gene-specific primers containing the T7 polymerase promoter sequence (Table 1) and a MEGAscript^TM^ RNAi Kit (Invitrogen). The quality and concentration of dsRNA were assessed using 1% agarose gel electrophoresis and a Thermo ultra-micro spectrophotometer, respectively. Based on the findings of P. E. Urwin, 1 mg/mL FITC solution was used as an indicator of dsRNA entry into *D. destructor* [48]. About 10,000 J3 worms were collected and immersed in a solution containing dsRNA and 3 mg/mL spermidine (Sigma, Shanghai, China), 50 mM octopamine (Sigma), and 5% gelatin. The worms were incubated in the solution with shaking at 120 rpm for 24 h at 25 °C in the dark. In addition, J3 worms that were incubated in green fluorescent protein (GFP) dsRNA served as the control. After 24 h of soaking in the solution, *D. destructor* specimens were washed three times with DEPC-water and immediately stored at −80 °C for later detection of gene expression or were collected into 1.5-mL centrifuge tubes for later detection of motility, chemotaxis, stylet thrusting, protein secretion, and reproduction coefficient.

### 4.7. Gene Expression Analysis after Gene Knockdown

To determine levels of *DdCα1D*, *DdCα1A,* and *DdCa_v_α_2_δ* gene transcription following gene knockdown, approximately 1000 nematodes were collected after soaking in dsRNA for 24 h. mRNA was extracted, and cDNA was synthesized as described above. Specific qPCR primers were designed using NCBI, and 18S rRNA was used as the internal control for all qPCR assays (Table 1). The qPCR reaction solution was a 20-μL mixture that included 1 μL cDNA template, 10 μL SYBR Green Premix Pro Taq HS qPCR mix (Accurate Biology, Changsha, China), 1 μL of each primer (0.2 mM), and 7 μL ddH_2_O. Quantitative analysis was performed using the 2^−ΔΔCT^ method, as described above.

### 4.8. Phenotypic Analysis after Gene Knockout

Post RNAi phenotype analysis was performed by assessing mobility, chemotaxis, stylet thrusting, protein secretion, and reproduction. 

*Ditylenchus destructor* mobility after the knockdown of *DdCα1D*, *DdCα1A* and *DdCa_v_α_2_δ* was assayed according to the procedure used by Kimber et al. [49]. One hundred treated J2s and control J2s were transferred into PVC tubes filled with moist sand, and the bottom of the PVC tubes were wrapped with 150-mesh nylon yarn. PVC tubes were transferred to a petri dish (50 mm) filled with 20 mL of ddH_2_O to cover the bottom of the sand column. Columns and petri dishes were kept in the dark at 25 °C. The number of worms passing through the sand column into the petri dish was counted at 6 h and 24 h, and the migration rating of the sand column was calculated. Migration rate (*Mr*) = *Pp* (passing population of nematodes)/100.

To determine whether chemotaxis was affected, we conducted experiments with 1-cm sweet potato blocks. Briefly, the blocks were placed on 1% agar in 90 mm petri dishes, 3 cm away from 200 J2s. Petri dishes were sealed with cling film and placed at 25 °C for 36 h. After incubation, the number of nematodes within 2 cm of the blocks was counted.

Worms were analyzed for stylet thrusting as previously reported by McClure et al., with slight modifications [50]. The concentrated suspension of aliquots (2 μL), containing approximately 50 J2 *D. destructor* specimens, was treated with 20 μL of 5 mM serotonin creatinine sulfate (Sigma) for 20 min, and 10 randomly selected J2s were observed for the frequency of stylet thrusting over a 1 min period. The number of twitches for each J2 in 60 s was calculated.

An assay using 0.1% resorcinol was developed to assess protein secretion in nematodes. Two thousand J2 nematodes from each treatment and control group were concentrated to 100 μL, an equal volume of 0.2% resorcinol was added, and specimens were incubated for 16 h at 25 °C. Lastly, the supernatant was aspirated, and the protein content was determined using a Modified BCA Protein Assay Kit (Sangon).

To assess whether the reproduction of nematodes treated with dsRNA was affected, 100 nematodes from each treatment group were inoculated into PDA medium filled with *Botrytis cinerea.* Then, 25 d after incubation at 25 °C, the total number of nematodes was counted, and the reproduction coefficient was calculated as the ratio of the final number of nematodes to the initial number of nematodes. Each assay experiment had three biological replicates and three technical replicates. Data were analyzed by one-way analysis of variance (ANOVA) in SPSS. Differences between treatments were tested using Duncan’s multiple range test (DMRT) with F test (DMRT) with *p* < 0.05 [51]. The significant difference letter marking method first ranked all the means from largest to smallest and then marked the largest mean with a; the average was compared with the following averages. If the difference was not significant (*p* > 0.05), the letter a was marked, and if the difference was significant (*p* < 0.05), the letter b was marked. If the mean was significantly (*p* < 0.05) less than the mean of the group marked by letter b, it was marked by letter c. 

## 5. Conclusions

This study identified the VGCC α_2_δ subunit in *D. destructor* and analyzed its transcriptional levels at different developmental stages and its tissue localization. In this study, silencing of the HVA α_1_ and α_2_δ subunits in *D. destructor* was achieved simultaneously using multi-targeted dsRNA soaking. The function of *DdCα1D,* as well as *DdCα1A,* was further validated, and the auxiliary role of *DdCa_v_α_2_δ* was demonstrated, enhancing our understanding of VGCCs in plant parasitic nematodes.

## Figures and Tables

**Figure 1 ijms-23-12999-f001:**
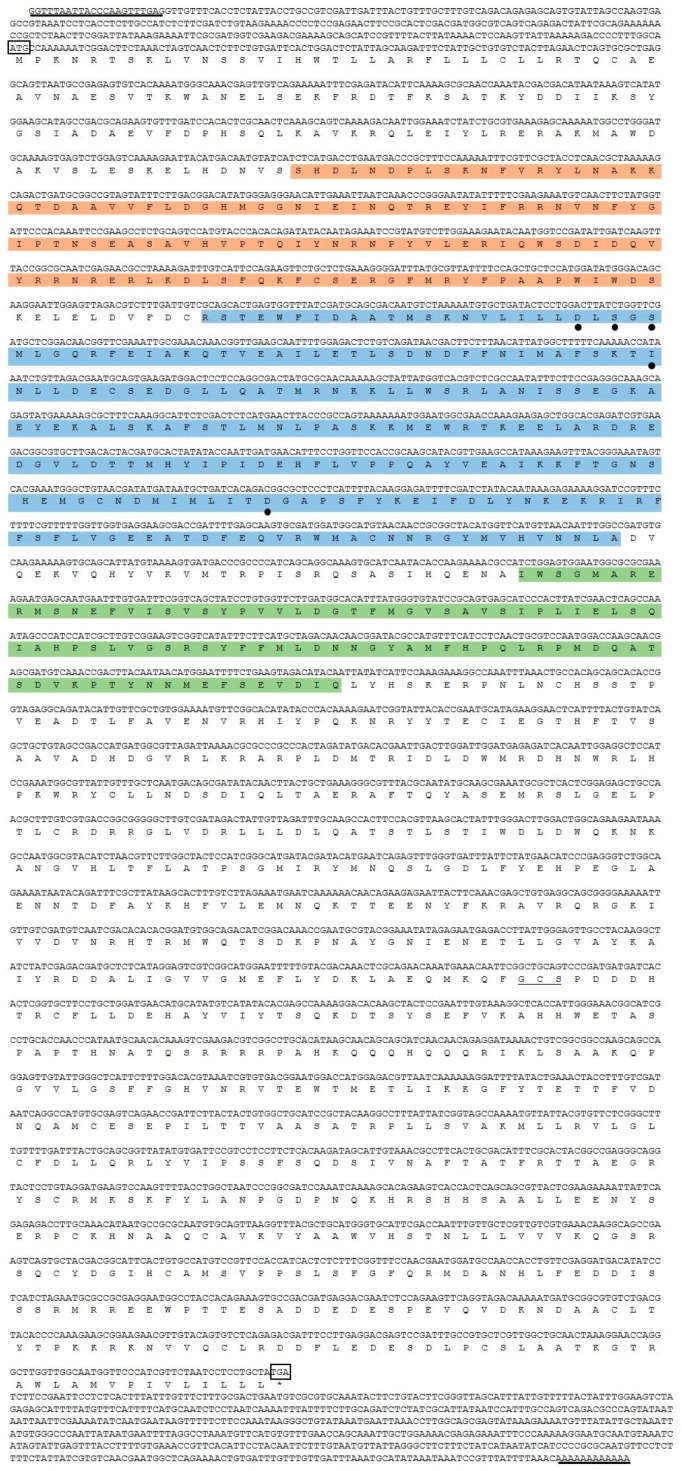
Sequence of *DdCa_v_α_2_δ* cDNA and its deduced amino acid sequence. The start codon (ATG) and the stop codon (TGA) are indicated by boxes. The nucleic acid sequence fragment before the ATG is the 5’UTR, and the nucleic acid sequence fragment after the TGA is the 3’UTR. The underlined region indicates the SL1 at the front of the 5’UTR and the polyA tail at the end of the 3’UTR. The asterisk (*) shows the stop codon (TGA). Orange shows the N-terminal of the VWA domain; blue and green show the VWA domain and Cache domain, respectively; the MIDAS motif is marked with a “•” symbol.

**Figure 2 ijms-23-12999-f002:**
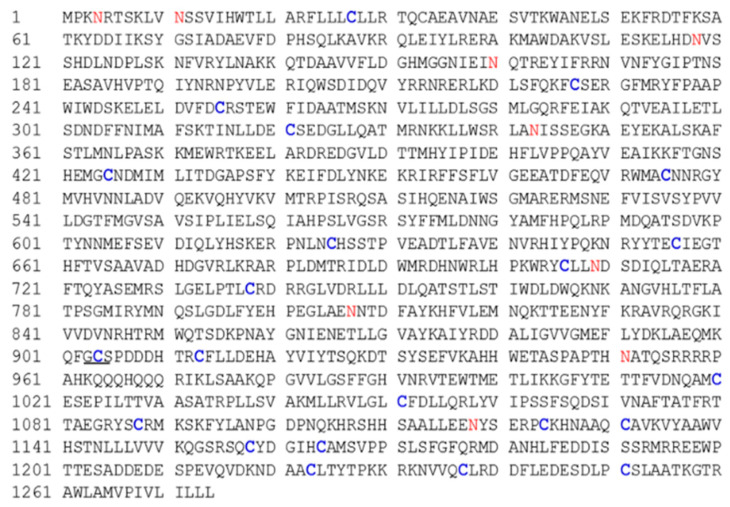
Amino acid sequence of the *DdCa_v_α_2_δ* subunit. Blue indicates cysteine residues, red indicates N-glycosylation sites, and underlining indicates suspected GPI anchor sites.

**Figure 3 ijms-23-12999-f003:**
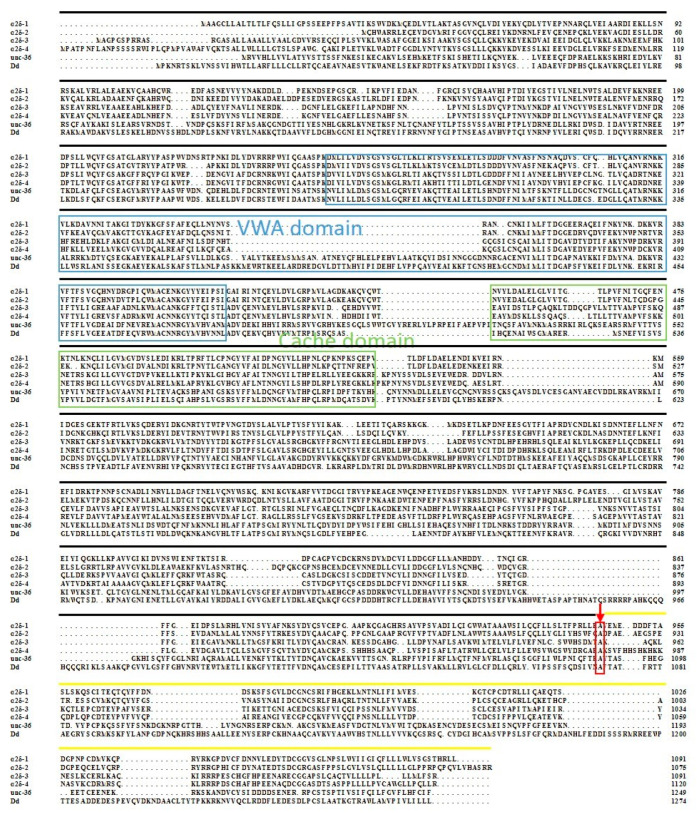
Multiple alignment analysis of *DdCa_v_α_2_δ* with other species’ Ca_v_α_2_δ. Blue indicates the VWA domain; green indicates the Cache domain; black indicates the α_2_; yellow indicates the δ; the red arrow indicates the breaking point between α_2_ and δ subunits. Other species’ Cavα_2_δ amino acid sequences from NCBI database (https://blast.ncbi.nlm.nih.gov, accessed on 15 October 2022), with the following accession numbers: human-α_2_δ-1 (NM_000722), human-α_2_δ-2 (AF042793), human-α_2_δ-3 (AF516696), human-α_2_δ-4 (AF516695), *unc-36* (NM_001047386.6).

**Figure 4 ijms-23-12999-f004:**
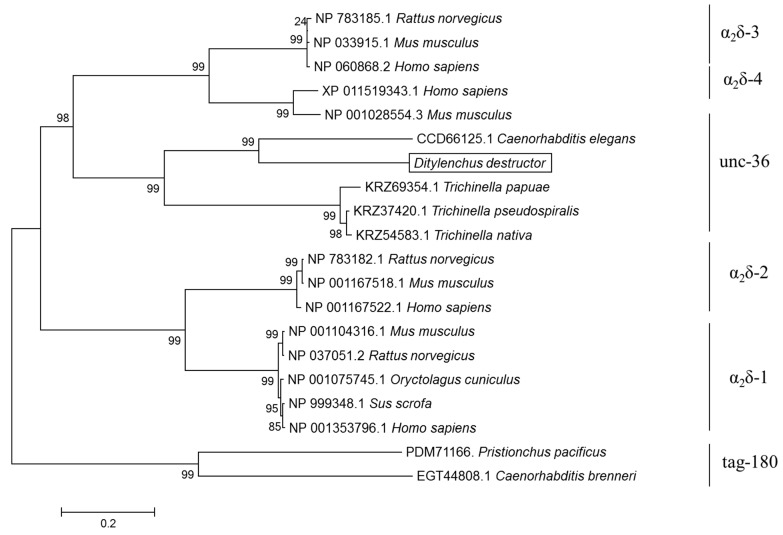
Phylogenetic analysis of *DdCa_v_α_2_δ* with α_2_δ genes from other species. The phylogenetic tree was constructed from the amino acid sequences of 20 VWA structural domain-containing α_2_δ genes. The tree was constructed using MEGA 6.0 based on the neighbor-joining method according to the amino acid sequences. The phylogenetic tree and sequences of other species were in the NCBI database; accession numbers are shown in Table 2.

**Figure 5 ijms-23-12999-f005:**
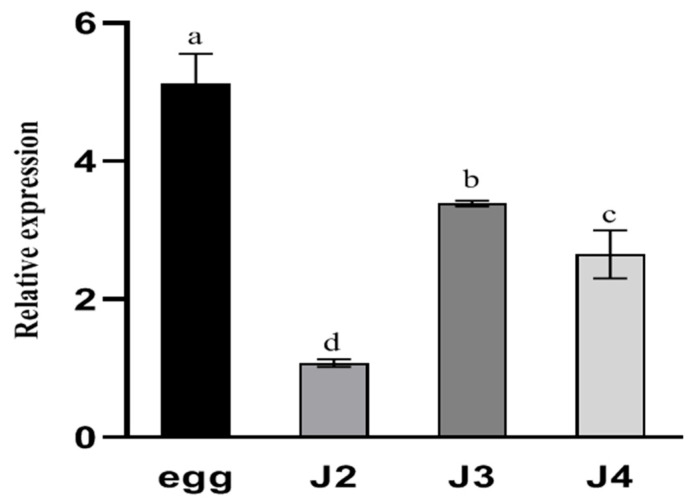
Relative transcript abundance of the *DdCa_v_α_2_δ* gene at each developmental stage. Using the transcript levels in J2 as a reference, *DdCa_v_α_2_δ* was significantly upregulated in other stages (J2, J3, and J4). Quantitative RT-PCR values are the mean ± standard error. Letters indicate significant differences according to Duncan’s one-way ANOVA of SPSS 21.0 Software (*p* < 0.05). Tissue localization analysis of *DdCa_v_α_2_δ* was conducted. The experiments were performed three times with similar results.

**Figure 6 ijms-23-12999-f006:**
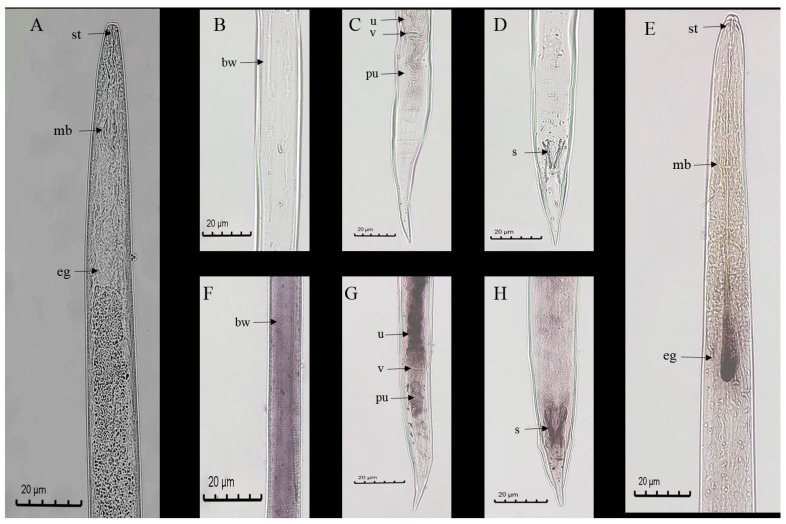
Tissue localization of *DdCa_v_α_2_δ* mRNA in the adult stages of *D. destructor.* (**A**–**D**): No hybridization signal was observed in *D. destructor* with the sense probe. (**E**–**H**): Tissue localization of *DdCa_v_α_2_δ* mRNA in *D. destructor* with the antisense probe. st: stylet; mb: median bulb; eg: esophageal glands; bw: body wall; u: uterus; v: vulva; pu: post uterine; s: spicules.

**Figure 7 ijms-23-12999-f007:**
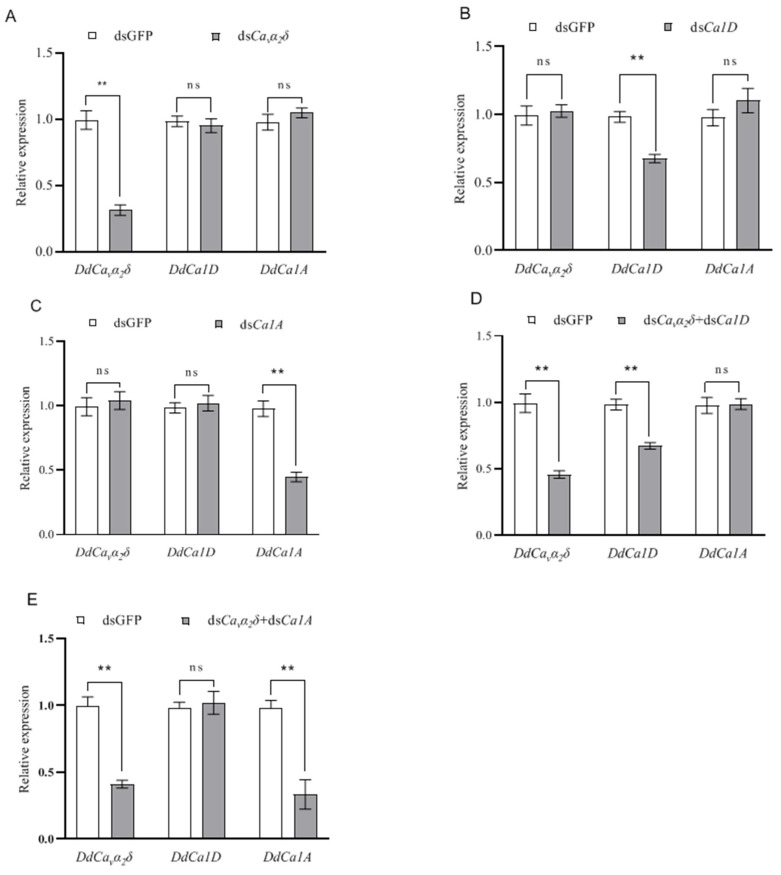
Effect of different dsRNA soaking solutions on the expression level of *DdCα1D*, *DdCα1A*, and *DdCa_v_α_2_δ* genes. Control nematodes were soaked in the solution containing non-target GFP dsRNA. (**A**) Nematodes treated with *DdCa_v_α_2_δ* dsRNA. (**B**) Nematodes treated with *DdCα1D* dsRNA. (**C**) Nematodes treated with *DdCα1A* dsRNA. (**D**) Nematodes treated with *DdCa_v_α_2_δ* and *DdCα1D* dsRNA. (**E**) Nematodes treated with *DdCa_v_α_2_δ* and *DdCα1A* dsRNA. After 24 h of dsRNA soaking, mRNA was extracted from *D. destructor* samples and then subjected to qRT-PCR analysis. Asterisks indicate significant differences based on Student’s *t* test, ** *p* < 0.01, ns indicates no significant difference. The experiments were performed three times with similar results.

**Figure 8 ijms-23-12999-f008:**
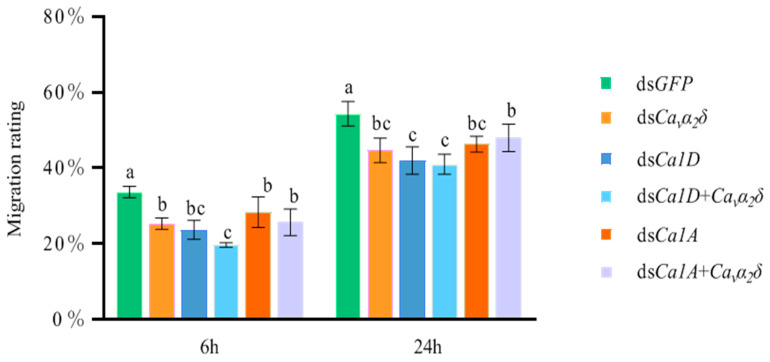
Effect of dsRNA soaking on the motility of *D. destructor*. One hundred J2-J3 worms treated with dsRNA were added to the sand column, and the number of worms passing through the sand column was counted at 6 h and 24 h, respectively, to calculate the migration rating. Each column represents the mean ± standard error of three replicates. Different letters indicate significant differences at *p* < 0.05 by Duncan’s multiple range test. The experiments were performed three times with similar results.

**Figure 9 ijms-23-12999-f009:**
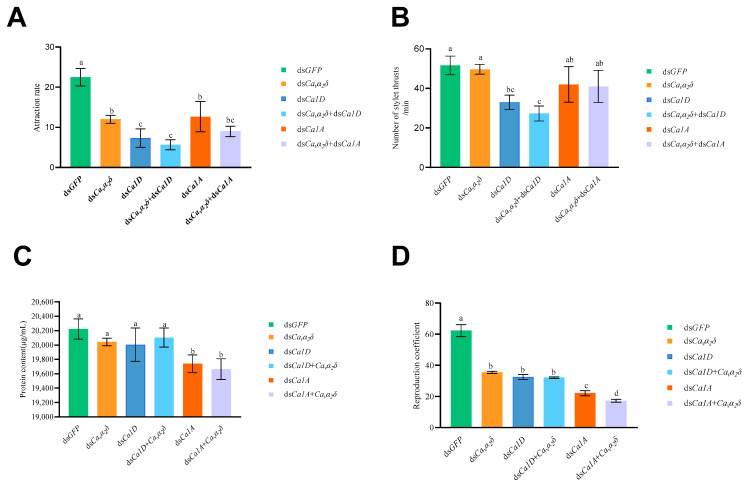
Effect of dsRNA soaking on the phenotype of *D. destructor*. (**A**) Effect of dsRNA soaking on the chemotaxis of *D. destructor*. (**B**) Effect of dsRNA soaking on the stylet thrusting of *D. destructor*. (**C**) Effect of dsRNA soaking on secreted proteins of *D. destructor.* (**D**) Effect of dsRNA soaking on the reproduction of *D. destructor*. The worms treated with clear water were set as the control. Different letters indicate significant differences at *p* < 0.05 by Duncan’s multiple range test. The experiments were performed three times with similar results.

**Table 1 ijms-23-12999-t001:** Primers used in the study.

Primer	Sequence (5′ to 3′)	Use
SL1	GGTTTAATTACCCAAGTTTGAG	Primers used for *DdCa_v_α_2_δ* cloning
5′-R1	ATGACAAATCTTTTAGGCGTTC	
5′-R2	GTGGGAATACCATAGAAGTTGAC	
D-F	AATGCCAAAAAATCGGACTTC	
D-R	ATAGCAGGAGGATTAGAACGATG	
3′ RACE outer primer	TACCGTCGTTCCACTAGTGATTT	
3′ RACE inner primer	CGCGGATCCTCCACTAGTGATTTCACTATAGG	
3′-F1	CCGTCCTCCTTCTCACAAGATAG	
3′-F2	TCCCATCGTTCTAATCCTCCTGCTA	
18S-F	CTGATTAGCGATTCTTACGGA	Primers for real-time PCR analysis
18S-R	AGAAGCATGCCACCTTTGA	
qα2δ-F	GCCTGGGATGCAAAAGTGAGTCA	
qα2δ-R	AGCGTTGAGGTAGCGAACGAAA	
qL-F	GACCCGTTATTGTTGAGCCA	
qL-R	ACGTTCCTTCGAGATGAGA	
qNL-F	TAGAAAACAGGCGAGACTTCC	
qNL-R	CTCATCCGTTGTTCGATCCTC	
HindIIIF	CCCAAGCTTGCTGCAGTCCCGATGATG	Primers for ISH analysis
EcoRI R	GGAATTCCCATGGTCCATCCGTCAC	
dsα_2_δ-F	TAATACGACTCACTATAGGGCAAGGCCTTTATTATCGGTAG	Primers used for synthesizing dsRNA
dsα_2_δ-R	TAATACGACTCACTATAGGGGTCGGCACTTTCTGTGGTAG	
dsL-F	TAATACGACTCACTATAGGGAGGAAGATGACCTCTTGTTAG	
dsL-R	TAATACGACTCACTATAGGGCCCAATATATGACCGTCTTTG	
dsNL-F	TAATACGACTCACTATAGGGCGCAACACGTACCAAACTC	
dsNL-R	TAATACGACTCACTATAGGGCTCATCTGAATCGCTAAGAGG	
dsGFP-F	TAATACGACTCACTATAGGGTACATCGCTCTTTCTTCACCG	
dsGFP-R	TAATACGACTCACTATAGGGACCAACAAGATGAAGAGCACC	

The underlined line indicates the restriction enzyme sites and the T7 promoter sequences.

**Table 2 ijms-23-12999-t002:** *DdCa_v_α_2_δ* information used for sequence alignment and phylogenetic analysis.

Species	Molecular Name/Accession Number	Identity (%)
*Caenorhabditis brenneri*	*tag-180*/EGT44808.1	22.0
*Caenorhabditis* *elegans*	*unc-36*/CCD66125.1	40.0
*Homo sapiens*	*alpha-2/delta-1*/NP_001353796.1	24.6
	*alpha-2/delta-2* /NP_001167522.1	24.3
	*alpha-2/delta-3*/NP_060868.2	27.0
	*alpha-2/delta-4*/XP_011519343.1	26.2
*Mus musculus*	*alpha-2/delta-1*/NP_001104316.1	24.4
	*alpha-2/delta-2*/NP_001167518.1	24.3
	*alpha-2/delta-3*/NP_033915.1	25.7
	*alpha-2/delta-4*/NP_001028554.3	25.5
*Oryctolagus cuniculus*	*alpha-2/delta-1*/NP_001075745.1	24.9
*Pristionchus pacificus*	*tag-180*/PDM71166.1	21.9
*Rattus norvegicus*	*alpha-2/delta-1*/NP_037051.2	24.5
	*alpha-2/delta-2* /NP_783182.1	23.3
	*alpha-2/delta-3* /NP_783185.1	27.3
*Sus scrofa*	*alpha-2/delta-1*/NP_999348.1	24.5
*Trichinella nativa*	*unc-36*/KRZ54583.1	31.3
*Trichinella papuae*	*unc-36*/KRZ69354.1	34.7
*Trichinella pseudospiralis*	*unc-36*/KRZ37420.1	31.3

## Data Availability

Not applicable.

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
