# Peer review of "The α2δ Calcium Channel Subunit Accessorily and Independently Affects the Biological Function of Ditylenchus destructor"

_ijms, 2022, doi:10.3390/ijms232112999_

Round 1

Reviewer 1 Report

Review report of the manuscript:

The α2δ calcium channel subunit accessorily and independently affects the biological function of Ditylenchus destructor

Authors: Xueling Chen, Mingwei An, Shan Ye, Zhuhong Yang  and Zhong Ding

Summary

The study investigated the function of α2δ subunit as one of the high voltage activated (HVA) calcium channels. In situ hybridization experiments showed that the DdCavα2δ subunit was expressed in the body wall, esophageal gland, uterus, post uterine sac and spicules. In addition, it was concluded that α2δ subunit has a role in the motility, reproduction, chemotaxis, stylet thrusting and protein secretion of the nematode.

General comments

The manuscript is clear, relevant for the field and mostly presented in a well-structured manner. The majority of the cited references are current. The manuscript is scientifically sound and the experimental design is appropriate to test the hypothesis.

Specific comments 

  • Please state how you have identified the nematode species?
  • Have you confirmed the nematode species identity by molecular method(s) and explain why?
  • Line 83. The beggining of the sentence should not be with abbreviation such as, D. destructor and throughout the text.
  • Figure 5. The genera and species names should be separated in some cases. The subphyla names should be in plural not in the singular form. The figure caption mentioned use of Mega 7 software, while in the Materials and Methods part it is written Mega 6.
  • Line 164. Phylogenic or phylogenetic analysis?
  • Figure 7. Please provide a better resolution/focus of the figure 7A so we can actually see the stylet and other internal organs.
  • Line 190. Medium build should be median bulb.
  • Lines: 333-387. The reference order is wrong.
  • Line 358-359. Shen et al. Requires the reference.
  • Lines: 408-505. The reference order is wrong.
  • Line 441. The software name requires the reference.
  • Line 492. Only the last name in the reference is usually used.

Author Response

Dear reviewer,

Thank you for constructive comments that have greatly improved our manuscript during this revision. We provide here a “point-to-point” response to address these comments. To discriminate the figures in the response letter and the manuscript, the figures shown in response to reviewers were named as FigR.+number. The reviewer’s comments are given in black font, and our response in red font. The changes described for Reviewers are presented in yellow highlight in the revised manuscript.

Review report of the manuscript:

The α2δ calcium channel subunit accessorily and independently affects the biological function of Ditylenchus destructor

Authors: Xueling Chen, Mingwei An, Shan Ye, Zhuhong Yang  and Zhong Ding

Summary

The study investigated the function of α2δ subunit as one of the high voltage activated (HVA) calcium channels. In situ hybridization experiments showed that the DdCavα2δ subunit was expressed in the body wall, esophageal gland, uterus, post uterine sac and spicules. In addition, it was concluded that α2δ subunit has a role in the motility, reproduction, chemotaxis, stylet thrusting and protein secretion of the nematode.

General comments

The manuscript is clear, relevant for the field and mostly presented in a well-structured manner. The majority of the cited references are current. The manuscript is scientifically sound and the experimental design is appropriate to test the hypothesis.

Response: We thank the reviewer for the positive comments on our work.

Specific comments 

  • Please state how you have identified the nematode species?Have you confirmed the nematode species identity by molecular method(s) and explain why?

Response: First, I used ITS primers( ITS-P1: 5’-CGTAACAAGGTAGCTGTAG-3’; ITS-P2: 5’-TTTCACTCGCCGTTACTAAGG-3’) to show that the nematodes used were stem nematodes (Figure 1). Furthermore, I used DdL1(5’-TTGTGTTTGCT GGTGCGCTTGT-3’)  and DdL2(to 5’- GAGTGAGAGCGATGTCAACATT -3’) show that it is B(L) type Ditylenchus destructor (Figure 2). And three samples were randomly selected for sequencing, and NCBI comparison showed Ditylenchus destructor. The pictures are as follows.

FigR. 1 Results of amplification of the internal transcribed spacer (ITS) universal primer. M: Mark 2000

FigR. 2 Type B (L) specific primer amplification results (252bp). M: Mark 2000

  • Line 83. The beggining of the sentence should not be with abbreviation such as, D. destructor and throughout the text.

Response: Thank you. We have corrected this in the revised MS on page 2, line 86 and have checked throughout the text.  

  • Figure 5. The genera and species names should be separated in some cases. The subphyla names should be in plural not in the singular form. The figure caption mentioned use of Mega 7 software, while in the Materials and Methods part it is written Mega 6.

Response:Thanks, we carefully re-do the evolutionary tree and have corrected mistakes in the revised manuscript in Figure 4.

  • Line 164. Phylogenic or phylogenetic analysis?

Response: Thank you. We have changed “ phylogenic analysis” to “ phylogenetic analysis” in the revised MS on page 7, lines 173.

  • Figure 7. Please provide a better resolution/focus of the figure 7A so we can actually see the stylet and other internal organs.

Response: Thank you for your constructive suggestions. We have changed the figure 7A in the revised MS on page 9. In addition, I have provided you with a picture of the structure of D. destructor from the nemaplex database (http://nemaplex.ucdavis.edu/Taxadata/G042s2.aspx) (Figure 3).

FigR. 3 Structure of D. destructor.

Line 190. Medium build should be median bulb.

Response: Thank you. We have changed “ medium build” to “ median bulb” in the revised MS on page 9, lines 197.

  • Lines: 333-387. The reference order is wrong.

Response: Thank you. We have changed this in the revised MS on lines 340-395.

  • Line 358-359. Shen et al. Requires the reference.

Response: Thank you. We have corrected this in the revised MS on page 14, line 365.

  • Lines: 408-505. The reference order is wrong.

Response: Thank you. We have changed this in the revised MS on lines 416-514.

  • Line 441. The software name requires the reference.

Response: Thank you. We have added reference in the revised MS on page 15, line 450.

  • Line 492. Only the last name in the reference is usually used.

Response: Thank you. We have corrected this mistake in the revised MS on page 16, line 501.

Reviewer 2 Report

In this manuscript the authors characterize the structure and the function of the gene encoding the a2d subunit of high voltage activated calcium channels in the plant parasitic nematode Ditylenchus destructor. To this end, they cloned the cDNA of the gene and they perform RNAi knock down experiments, targeting various subunits of voltage gated calcium channels. I would like to point out that I am rather a specialist for nematodes than for calcium channels. Therefore, I cannot evaluate, if all the background information provided is correct (I have no reason to believe that it is not). In principle, this is a very interesting study on an economically important plant pathogen. However, I do have a number of rather serious issues with the experiments and their presentation and interpretation. At places the authors tend to draw firm conclusions from very small effects. Unfortunately, the manuscript was not very carefully prepared. Some essential information about the experimental procedures and the statistical analysis is missing and while one section is duplicated there is obviously a piece of text missing (see specific points). In its present form, this manuscript is not acceptable for publication.

Below I provide my comments ordered according to the paragraphs in the result section. These comments include also the corresponding sections in Materials and Methods

Specific points:

Introduction (all points are minor):

Line 52: a2d consists of the two subunits a2 and d (two proteins) but later there is one gene mentioned for both of them. Please clarify here that a2 and d are two proteins encoded by the same locus (is this correct?).

Lines 80 - 85: Please provide the phylogenetic position of D. destructor within the nematodes. To which of the major clades does it belong?

Lines 87,88: It would be helpful to provide the information given in lines 358-364 here already.

Line 96: Why "Conversely"?

2.1. Cloning and characterizing of DdCava2d

The arguments for the identified gene being the true and only a2d encoding gene in D. destructor should be made more clear. Is there a full genome sequence for D. destructor? Might there be a second paralog in this species? Are the identified gene and C. elegans unc-36 best reciprocal BLAST hits? Is there a tag-180 like gene?

Line 107 (minor): To my understanding the SL1 sequence has been used as primer site to amplify the cDNA. Therefore, the fact that this worked but not sequence analysis demonstrated the presence of SL1.

Figures 2,3,4 (minor): In order to make the connection to the introduction it would be helpful to indicate what corresponds to a2 and what to d.

Figure 3 (minor): I think this figure is not necessary.

Table 2 (minor): This table is not mentioned in the text. I assume, it is supposed to lists all the sequences used in Figure 5. If this is the case, C. elegans tag-180 is missing.

Figure 5 (minor): In several cases the spaces between the genus and the species names are missing (Homosapiens instead of Homo sapiens).

2.3. Developmental expression of DdCava2d

These experiments are not described in sufficient detail. Was there any kind of control for possible DNA contamination (control reactions without reverse transcriptase, primers spanning an intron...)? I assume that "All experiments were repeated three times" means that the developmental stages were isolated and their RNA analyzed three times independently (three biological replicates). How many technical replicates were done per biological replicate? How were the technical and biological replicates averaged? How much RNA was obtained from the 1000 worms and how much of it was used for reverse transcription and how much of this for the qPCR? In what range were the Ct values for DdCava2d  and for 18S? Is 18S RNA really a suitable reference for RNA that was isolated using a poly(A) enrichment protocol?

Line 447: "Table1" not "Table S1".

Section 2.4.

This section is a literal repetition of 2.3. A substantial portion of text  describing the experiments illustrated in Figure 7 appears to be missing.

Figure 7: Which developmental stage is shown?

2.5. Silencing efficiency of dsRNA soaking

How many replicates were done for these experiments?

Some of the comments to section 2.3 apply also to these qPCR experiments.

Line 208: To claim that the genes were "effectively silenced" is a dramatic exaggeration. The effect described is moderate at best.

Line 213,214: From the fact that the two genes do not appear to influence each other's expression (at the RNA level) it cannot be concluded that they function independently. It is interesting, however, that with this respect the D. destructor genes appear to behave differently from their mammalian counterparts. This could be mentioned in the discussion.

Line 216 (minor): This figure shows the effect on the steady state level of the transcripts, not the on the transcription. Presumably the RNAi reduces the stability of the mRNAs, not their synthesis.

Here and throughout the manuscript (minor): "a" and "a" are used interchangeably in gene/protein names (for example sometimes it is Ca1D, sometimes Ca1D).

2.6.1 Motility assay

In the caption to Figure 9 it is stated that "Different letters indicate significant differences..." but it is not explained which letter stands for which comparison. I can therefore not fully understand and evaluate the figure. However, I do not think that the conclusion about the auxiliary role of DdCava2d in lines 243,244 can be drawn based on these results.

The mentioned "Duncan's new complex difference method" is neither described nor is there a reference provided.

Figure 10 (includes data belonging to sections 2.6.2 - 2.6.5):

Again, in the caption it is stated that "Different letters indicate significant differences..." but it is not explained which letter stands for which comparison. Neither here nor in Materials and Methods it is stated how many replicates were done. It is therefore very difficult to evaluate the sections 2.6.2 to 2.6.5

2.6.2 Chemotaxis assay

Might the effect on motility have influenced the outcome of the chemotaxis assay because some worms were just not capable of reaching the target due to their impaired locomotion?

2.6.2 Stylet thrusting assay

Lines 285-287: This appears to be a contradiction in itself. What do the authors want to say here?

2.6.4 Protein secretion assay

To me all bars in figure 10C look the same. The authors make claims based on a measured reduction of protein secretion of less than 3%. Can these protein concentrations really be measured so accurately? Even if these differences do appear significant in some statistical test (was there a correction for multiple testing?) I do not think that the observed differences are biologically relevant and large enough to draw conclusions.

2.6.5. Reproduction assay

Line 320-322: This conclusion is again based on a minute difference between the Ca1D and the Cava2d  Ca1D double knock down (32.4 versus 31.5). Given the unavoidable noise and possible small systematic errors in this kind of experiments, I do not believe that this difference is really biologically significant.

Discussion and Conclusions

These two sections will need to be modified extensively following the revision of the result section. Therefore, I do not comment in detail on the current version.

Author Response

Dear reviewer,

Thank you for constructive comments that have greatly improved our manuscript during this revision. We provide here a “point-to-point” response to address these comments. The reviewer’s comments are given in black font, and our response in red font. The changes described for Reviewers are presented in yellow highlight in the revised manuscript.

Comments and Suggestions for Authors

 In this manuscript the authors characterize the structure and the function of the gene encoding the a2d subunit of high voltage activated calcium channels in the plant parasitic nematode Ditylenchus destructor. To this end, they cloned the cDNA of the gene and they perform RNAi knock down experiments, targeting various subunits of voltage gated calcium channels. I would like to point out that I am rather a specialist for nematodes than for calcium channels. Therefore, I cannot evaluate, if all the background information provided is correct (I have no reason to believe that it is not). In principle, this is a very interesting study on an economically important plant pathogen. However, I do have a number of rather serious issues with the experiments and their presentation and interpretation. At places the authors tend to draw firm conclusions from very small effects. Unfortunately, the manuscript was not very carefully prepared. Some essential information about the experimental procedures and the statistical analysis is missing and while one section is duplicated there is obviously a piece of text missing (see specific points). In its present form, this manuscript is not acceptable for publication. 

Response: Thank you for your comments. We have responded to the following questions one by one and/or made corresponding changes in the MS.

Below I provide my comments ordered according to the paragraphs in the result section. These comments include also the corresponding sections in Materials and Methods

 Specific points:

 Introduction (all points are minor):

 Line 52: a2d consists of the two subunits a2 and d (two proteins) but later there is one gene mentioned for both of them. Please clarify here that a2 and d are two proteins encoded by the same locus (is this correct?).

Response: Yes, a2d consists of two subunits, a2 and d, and both subunits are the product of a single gene. Structurally, the α2δ subunit is a heavily glycosylated 175 kDa protein that is post translationally cleaved to yield a disulfide-linked α2 and δ protein (Norbert et al., 1999). We have rewritten this sentence in the revised MS on page 2, lines 52-53.

Lines 80 - 85: Please provide the phylogenetic position of D. destructor within the nematodes. To which of the major clades does it belong?

 Response: Thank you for your comments. According to small subunit ribosomal DNA (SSU rDNA) sequences tree, Ditylenchus destructor located at clades 12 (Holovachov et al., 2009). We have added this in the revised MS on page 2, lines 81-83.

Lines 87,88: It would be helpful to provide the information given in lines 358-364 here already.

Response: Thank you for your constructive suggestions. We altered the sentence on page 2, line 89-93.

Line 96: Why "Conversely"?

Response:Sorry! It is our mistake. We have changed “Conversely” to “Consequently” in the revised MS on page 3, lines 100.

 2.1. Cloning and characterizing of DdCava2d

 The arguments for the identified gene being the true and only a2d encoding gene in D. destructor should be made more clear. Is there a full genome sequence for D. destructor? Might there be a second paralog in this species? Are the identified gene and C. elegans unc-36 best reciprocal BLAST hits? Is there a tag-180 like gene?

 Response: We blasted the cDNA sequences of DdCavα2δ at WormBase (https://parasite.wormbase.org/index.html) databases and found no tag180-like genes in the D. destructor genome.

Line 107 (minor): To my understanding the SL1 sequence has been used as primer site to amplify the cDNA. Therefore, the fact that this worked but not sequence analysis demonstrated the presence of SL1.

Response:Thank you for your expertise comments. SL1 sequence not only is a primer, but also is a guide sequence present at the 5' end of the mRNA of most nematodes (Blaxter and Liu, 1996). We have added it the revised MS on page 3, lines 112-113.

Figures 2,3,4 (minor): In order to make the connection to the introduction it would be helpful to indicate what corresponds to a2 and what to d.

Response:Thank you for your good suggestions. We have added it in Figures 3 in the revised MS.

Figure 3 (minor): I think this figure is not necessary.

Response:As suggested. We deleted the Figure in the revised MS.

Table 2 (minor): This table is not mentioned in the text. I assume, it is supposed to lists all the sequences used in Figure 5. If this is the case, C. elegans tag-180 is missing.

Response:Thanks. The Table 2 has mentioned on lines 167. And We have changed Figure 4 and Table 2 in the revised MS.

Figure 5 (minor): In several cases the spaces between the genus and the species names are missing (Homosapiens instead of Homo sapiens).

Response:Thanks, we carefully re-do the evolutionary tree and have corrected mistakes in the revised manuscript in Figure 4 in the revised MS.

 2.3. Developmental expression of DdCava2d

 These experiments are not described in sufficient detail. Was there any kind of control for possible DNA contamination (control reactions without reverse transcriptase, primers spanning an intron...)? I assume that "All experiments were repeated three times" means that the developmental stages were isolated and their RNA analyzed three times independently (three biological replicates). How many technical replicates were done per biological replicate? How were the technical and biological replicates averaged? How much RNA was obtained from the 1000 worms and how much of it was used for reverse transcription and how much of this for the qPCR? In what range were the Ct values for DdCava2d and for 18S? Is 18S RNA really a suitable reference for RNA that was isolated using a poly(A) enrichment protocol?

Response: Each experiment had three independent biological replicates, each biological replicate was performed with three technical replicates. We used 15 μl of RNase free water to dissolve RNA extracted from 1000 worms, 11 μl for cDNA synthesis. The total volume of qPCR reaction fluid per tube is 25μl, and the volume of cDNA is 2μl. The Ct values of qPCR were all between 18 and 35, which is plausible.

Line 447: "Table1" not "Table S1".

Response:Thanks, we altered it on page 15, line 441.

Section 2.4.

 This section is a literal repetition of 2.3. A substantial portion of text describing the experiments illustrated in Figure 7 appears to be missing.

 Response:Thank you for your rigorous review. We have removed duplicate statements and added the description in Figure 6 in the revised MS on page 8, line 188-193.

Figure 7: Which developmental stage is shown?

Response: Thanks. In situ hybridization, approximately 10,000 mixed-stage, including J2 to adult, D. destructor was used to detect the site of DdCavα2δ gene expression in D. destructor at the mRNA level. We have described it on page 15, line 465.

 2.5. Silencing efficiency of dsRNA soaking

 How many replicates were done for these experiments?

 Some of the comments to section 2.3 apply also to these qPCR experiments.

 Response: Thanks! In fact, our experiments have ≥ 3 biological replicates. We illustrated this in the revised MS on page 8, line 187 and page 10, line 231.

Line 208: To claim that the genes were "effectively silenced" is a dramatic exaggeration. The effect described is moderate at best.

Response:Thanks. We have changed “effectively silenced” to “a certain degree of silenced” in the revised MS on page 9, lines 215.

Line 213,214: From the fact that the two genes do not appear to influence each other's expression (at the RNA level) it cannot be concluded that they function independently. It is interesting, however, that with this respect the D. destructor genes appear to behave differently from their mammalian counterparts. This could be mentioned in the discussion.

Response:Thank you for your rigorous suggestion, we have changed in the revised MS on page 9, lines 218-222 and added it in the discussion on page 14, lines 382-385.

Line 216 (minor): This figure shows the effect on the steady state level of the transcripts, not the on the transcription. Presumably the RNAi reduces the stability of the mRNAs, not their synthesis.

 Response: Thank you. We agree with reviewer’s comment. We have changed “transcription” to “expression level” in the revised MS on page 10, lines 224.

Here and throughout the manuscript (minor): "a" and "a" are used interchangeably in gene/protein names (for example sometimes it is Ca1D, sometimes Ca1D).

Response: Thank you. We have corrected this in the revised MS and have checked throughout the text.

 2.6.1 Motility assay

 In the caption to Figure 9 it is stated that "Different letters indicate significant differences..." but it is not explained which letter stands for which comparison. I can therefore not fully understand and evaluate the figure. However, I do not think that the conclusion about the auxiliary role of DdCava2d in lines 243,244 can be drawn based on these results.

 Response: Thank you. The significant difference letter marking method first ranks all the means from largest to smallest, and then marks the largest mean with a; the average is compared with the following averages. If the difference is not significant (P > 0.05), mark the letter a, and if the difference is significant (P < 0.05), mark the letter b. Meanwhile, we have changed the conclusion about the auxiliary role of DdCava2d  on page 11, lines 250-253.

The mentioned "Duncan's new complex difference method" is neither described nor is there a reference provided.

Response: Thank you. This way was named “Duncan's multiple range test”, I have added the reference on page 17, lines 531.

Figure 10 (includes data belonging to sections 2.6.2 - 2.6.5):

 Again, in the caption it is stated that "Different letters indicate significant differences..." but it is not explained which letter stands for which comparison. Neither here nor in Materials and Methods it is stated how many replicates were done. It is therefore very difficult to evaluate the sections 2.6.2 to 2.6.5

Response: Thanks. The significant difference letter marking method see above. Moreover, I have stated that each assay experiment has three biological replicates and three technical replicates in the revised MS in lines 528-529.

2.6.2 Chemotaxis assay

 Might the effect on motility have influenced the outcome of the chemotaxis assay because some worms were just not capable of reaching the target due to their impaired locomotion?

Response: Impaired motility in nematodes affects chemotaxis. Considering the injury of spermidine agent to the motility of nematodes, we treated nematodes with spermidine agent plus GFP as control. 

2.6.2 Stylet thrusting assay

 Lines 285-287: This appears to be a contradiction in itself. What do the authors want to say here?

 Response: Thank you. We have rewritten this sentence in the revised MS on page 12, lines 294-296.

2.6.4 Protein secretion assay

 To me all bars in figure 10C look the same. The authors make claims based on a measured reduction of protein secretion of less than 3%. Can these protein concentrations really be measured so accurately? Even if these differences do appear significant in some statistical test (was there a correction for multiple testing?) I do not think that the observed differences are biologically relevant and large enough to draw conclusions.

 Response: Sorry, it was our mistake! we have changed to a more suitable unit in the revised MS on page 12, lines 300-307. Resorcinol can lead to protein secretion in many tissues and organs of nematodes, such as body wall, head sensory organ, tail sensory organ and esophageal gland. However, RNAi only specifically acts on the corresponding genes in this experiment, so the influence is limited. Under the stimulation of resorcinol, the secretion base of nematodes itself is higher, but the corresponding treatments are statistically different.

2.6.5. Reproduction assay

 Line 320-322: This conclusion is again based on a minute difference between the Ca1D and the Cava2d  Ca1D double knock down (32.4 versus 31.5). Given the unavoidable noise and possible small systematic errors in this kind of experiments, I do not believe that this difference is really biologically significant.

Response: Thank you. We revised this conclusion in the revised MS on page 12, lines 328-330.

 Discussion and Conclusions

 These two sections will need to be modified extensively following the revision of the result section. Therefore, I do not comment in detail on the current version.

Response: Thank you. The discussion and conclusion of our new version have been carefully revised.

References

Blaxter M, Liu L. Nematode spliced leaders--ubiquity, evolution and utility. Int J Parasitol, 1996 ,26,1025-33. PMID: 8982784.

Klugbauer N, Lacinová L, Marais E, Hobom M, Hofmann F. Molecular diversity of the calcium channel alpha2delta subunit. J Neurosci, 1999,19,684-91. doi: 10.1523/JNEUROSCI.19-02-00684.1999.

Holovachov, Oleksandr, van Megen, Hanny Bongers T; Bakker J, Helder J, van den Elsen S, Holterman M, Karssen G, Mooyman, Paul. A phylogenetic tree of nematodes based on about 1200 full-length small subunit ribosomal DNA sequences.  Nematology, 2009,11, 927–950.

Round 2

Reviewer 2 Report

The revised manuscript is much improved and the authors addressed most of the points I had raised to my satisfaction. Most importantly, they clarified that indeed for all experiments multiple independent biological replicates have been done. have been done.

I do have a few remaining points that, however, are rather minor and the authors can be trusted to take care of in a found of minor revision without re-review by the reviewers.

Specific points:

Line 110,111 (response to my comment to line 107 of the original manuscript): Here I was misunderstood. I think it is not correct to say that sequence analysis demonstrated that an SL1 was present. The PCR reaction was performed using the SL1 sequence as primer. Therefore, this sequence is derived from the primer. However, the fact that this primer worked to amplify the cDNA shows that the original mRNA had an SL1 at its 5' end and that the cDNA the authors isolated extends to the real 5' end of the mRNA. The sentence the authors added is also useful. However, SL1 is present at the 5' end of many (not all) mRNAs f most nematodes.

Line 172: the information is in Table 2. reproduced right below the figure in my version of the manuscript, not supplementary table 2. Or, is the table intended to be supplementary material? Them the Table should be labelled as Supplementary Table 2.

Lines 188 - 202: Please indicate which developmental stage is shown in Figure 6.

Lines 218 - 220: I suggest rephrasing to something like: "...fed, this led to a significant specific partial knock down of the target gene, indicating that our RNAi approach works."

Line 224 first word: "does" not "is".

Legends to figures 8 and 9: I still think, the letters indicating significant differences are not sufficiently explained. The authors provide some explanation in their response but not in the manuscript. However, in their response only the letters a and b, but not c, are explained. In the legend it needs to be stated from which other samples something labelled with b or c is significantly different.

Lines 386 - 389: I do not really understand this point here? I do not think that from the mentioned RNAi results it can be concluded that in D. destructor the α2δ subunit does not interact with the α1 subunit and increase the amount of α1 subunit protein associated with the plasma membrane, as in mammals.

I guess the comment was included because of my remark to lines 213,241 in the original manuscript. However, there it was about the RNA levels and I had understood the introduction such that in other systems knocking down one of the genes in question led to a down regulation of another gene at the RNA level. I think this discussion point is not essential and I would just omit it.

Author Response

Dear reviewer,

Thank you for your significant comments, which very helpful for the revision and improvement of our manuscript during this revision. We have studied comments carefully and have made correction which we hope meet with approval. We provide here a “point-point” response to address these comments. The reviewer’s comments are given in black font, and our response in red font. The corresponding modifications in the text have been marked in yellow highlight.

Comments and Suggestions for Authors

The revised manuscript is much improved and the authors addressed most of the points I had raised to my satisfaction. Most importantly, they clarified that indeed for all experiments multiple independent biological replicates have been done.

I do have a few remaining points that, however, are rather minor and the authors can be trusted to take care of in a found of minor revision without re-review by the reviewers.

Response: Thanks especially for your positive comments. We have responded to the following questions one by one and/or made corresponding changes in the MS.

Specific points:

Line 110,111 (response to my comment to line 107 of the original manuscript): Here I was misunderstood. I think it is not correct to say that sequence analysis demonstrated that an SL1 was present. The PCR reaction was performed using the SL1 sequence as primer. Therefore, this sequence is derived from the primer. However, the fact that this primer worked to amplify the cDNA shows that the original mRNA had an SL1 at its 5' end and that the cDNA the authors isolated extends to the real 5' end of the mRNA. The sentence the authors added is also useful. However, SL1 is present at the 5' end of many (not all) mRNAs f most nematodes.

Response: Thanks. The presence of 5’ trans -spliced leader sequences in nematodes allows the use of an SL1-PCR strategy to clone full-length cDNAs from very small amounts of RNA (Blaxter et al., 1996; Mitreva et al., 2004). The DdCavα2δ sequence obtained by SL1-PCR and 3’RACE strategy has a polyA tail at its 3' end. Therefore, we indicating that this is the complete cDNA sequence. And, we have added these sentences in the revised MS on page 3, line 111-118.

Line 172: the information is in Table 2. reproduced right below the figure in my version of the manuscript, not supplementary table 2. Or, is the table intended to be supplementary material? Them the Table should be labelled as Supplementary Table 2.

Response: Thank you. We have changed “in the Supplementary Table 2” to “in the Table 2” in the revised MS on page 7, lines 176-177.

Lines 188 - 202: Please indicate which developmental stage is shown in Figure 6.

Response: Thanks. We have indicated that the adult stages of D. destructor in Figure 6 on page 8, line 194-195.

Lines 218 - 220: I suggest rephrasing to something like: "...fed, this led to a significant specific partial knock down of the target gene, indicating that our RNAi approach works."

Response: Thank you for your constructive suggestions. We corrected the sentence on page 9, line 224-226.

Line 224 first word: "does" not "is".

Response: Sorry! It is our mistake. We have changed “is” to “does” in the revised MS on page 9, lines 230.

Legends to figures 8 and 9: I still think, the letters indicating significant differences are not sufficiently explained. The authors provide some explanation in their response but not in the manuscript. However, in their response only the letters a and b, but not c, are explained. In the legend it needs to be stated from which other samples something labelled with b or c is significantly different.

Response: Thank you. The significant difference letter marking method first ranks all the means from largest to smallest, and then marks the largest mean with a; the average is compared with the following averages. If the difference is not significant (P>0.05), mark the letter a, and if the difference is significant (P < 0.05), mark the letter b. If the mean is less than the mean marked by the letter b and the difference is significant (P < 0.05), it is indicated by the letter c. Meanwhile, we have added the statement in the revised MS on page 17, lines 545-550.

Lines 386 - 389: I do not really understand this point here? I do not think that from the mentioned RNAi results it can be concluded that in D. destructor the α2δ subunit does not interact with the αsubunit and increase the amount of α1 subunit protein associated with the plasma membrane, as in mammals. I guess the comment was included because of my remark to lines 213,241 in the original manuscript. However, there it was about the RNA levels and I had understood the introduction such that in other systems knocking down one of the genes in question led to a down regulation of another gene at the RNA level. I think this discussion point is not essential and I would just omit it.

Response: Thank you for your comment. We deleted this sentence in the revised manuscript.

References

Blaxter M, Liu L. Nematode spliced leaders--ubiquity, evolution and utility. Int J Parasitol, 1996 ,26,1025-33. PMID: 8982784.

Mitreva M, Elling AA, Dante M, Kloek AP, Kalyanaraman A, Aluru S, Clifton SW, Bird DM, Baum TJ, McCarter JP. A survey of SL1-spliced transcripts from the root-lesion nematode Pratylenchus penetrans. Mol Genet Genomics, 2004,272,138-48. doi: 10.1007/s00438-004-1054-0.
